# UV Treatment Improves the Biocompatibility and Antibacterial Properties of Crystallized Nanostructured Titanium Surface

**DOI:** 10.3390/ijms20235991

**Published:** 2019-11-28

**Authors:** Mai Hatoko, Satoshi Komasa, Honghao Zhang, Tohru Sekino, Joji Okazaki

**Affiliations:** 1Department of Removable Prosthodontics and Occlusion, Osaka Dental University, 8-1, Kuzuhahanazono-cho, Hirakata-shi, Osaka 573-1121, Japan; hatoko-m@cc.osaka-dent.ac.jp (M.H.); joecheung-asuka@hotmail.com (H.Z.); joji@cc.osaka-dent.ac.jp (J.O.); 2The Institute of Scientific and Industrial Research, Osaka University, Suita, Osaka 565-0871, Japan; sekino@sanken.osaka-u.ac.jp

**Keywords:** UV treatment, crystallization, implant, bone differentiation, antibacterial

## Abstract

This study describes the production of a new material composed of pure titanium (Ti) metal with a crystallized nanostructure and investigated whether heat treatment and ultraviolet (UV) irradiation improved its biocompatibility and antibacterial properties. We compared the performance of UV-irradiated and non-irradiated Ti nanosheets (TNS) formed by dark alkaline treatment and heating at 600 °C with that of untreated pure Ti nanostructure (positive control). In vitro and in vivo experiments to assess biocompatibility and effects on cell behavior were performed using human umbilical vein endothelial cells and rat bone marrow cells. The material surface was characterized by X-ray photoelectron spectroscopy (XPS). The antibacterial properties of the irradiated material were evaluated using *Staphylococcus aureus*, a common pathogenic bacterium. The UV-irradiated TNS exhibited high angiogenic capacity and promoted cell adherence and differentiation relative to the control. Further, surface analysis via XPS revealed a lower C peak for the UV-treated material, indicating a reduced amount of dirt on the material surface. Moreover, UV irradiation decreased the viability of *S. aureus* on the material surface by stimulating reactive oxygen species production. The biocompatibility and antibacterial properties of the TNS were improved by UV irradiation. Thus, TNS may serve as a useful material for fabrication of dental implants.

## 1. Introduction

Titanium (Ti) is a useful element in clinical dentistry owing to its excellent mechanical properties and biocompatibility [1,2,3,4]. Since it cannot directly adhere to bone, the Ti surface is usually modified to enhance bone formation around the implant as, in general, rough surfaces can stimulate cell attachment, differentiation, and formation of extraosseous matrix [5,6,7,8,9]. Recent advances in dental implant research have enabled modification of the surface of implant materials at the nanometer scale [10,11,12,13,14,15]. Such nanostructures are biomimetic—i.e., they reproduce the natural cellular environment—and thus, can directly affect the behavior of osteogenic cells near the implant. However, although the capacity of implants to induce hard tissue differentiation is constantly improving, the risk of peri-implant inflammation due to bacterial adhesion to the surface remains a significant issue [16,17,18,19]. Indeed, we previously showed that pure Ti metal nanostructures formed by concentrated alkali treatment promoted the adhesion of inflammation-causing bacteria compared with an untreated pure Ti surface [20]. Therefore, a need exists for the development of an implant material that is not only biocompatible, but also has antimicrobial properties.

Ti dioxide (TiO_2_), also known as titania, is a naturally occurring oxide of Ti that has been used as a photocatalyst for the decomposition of organic compounds [21]. The oxide film on the Ti surface generally exists in an amorphous state and does not exhibit photocatalytic activity. However, the three crystal structures of TiO_2_—i.e., anatase, rutile, and brookite—can be rendered photocatalytically reactive by various oxidation methods. It is generally acknowledged that anatase titania is a more efficient photocatalyst than rutile titania [22]. Thus, it is possible to develop a material from Ti, crystallized with anatase, that can effectively induce hard tissue differentiation while inhibiting bacterial colonization.

A Ti nanosheet (TNS) is a nanostructure similar to TiO_2_ (titania) nanotubes created by the deposition of Ti via TiO_2_ sputtering [23]. Titania nanotubes and TNSs can be formed on Ti metal surfaces by 10 M aqueous NaOH treatment at 30 °C, which produces nanoscale roughness. TNSs produced by chemical treatment were shown to promote osteogenic differentiation around implant tissue both in vitro and in vivo [24,25,26,27,28,29,30]. Material structure and surface properties play important roles in protein adsorption, which can affect cell behavior. Application of amelogenin—which readily adsorbs proteins—to a TNS resulted in the formation of only hard tissues around the implants [31]. Further, a TNS structure was successfully crystallized by heat treatment at 600 °C after soaking pure Ti metal and Ti alloy in an aqueous solution of NaOH for 24 h [32]. In a recent study, we examined how the surface of TNS modified Ti, and how the Ti4Al6V alloy is affected by heat treatment at multiple temperatures. As a result, heat treatment at 600 °C formed a TNS crystal structure in which an amorphous layer of alkali titanate was firmly bonded to the substrate on the surface of Ti and the Ti4Al6V alloy. Ti surface of this crystallized TNS showed higher capacity to induce hard tissue differentiation of rat bone marrow (RBM) cells than did a TNS with a pure Ti metal surface. Additionally, pure Ti deposited on a TNS and irradiated with ultraviolet (UV) light showed strong antibacterial activity while maintaining the capacity to induce hard tissue differentiation [20]. Hence, these are useful properties to promote periodontal tissue regeneration for a material used in implants. To date, we have analyzed various data concerning the biocompatibility of nanostructured pure titanium metal. However, these previous studies have been shown to be specific to an individual variable such as heat treatment, UV treatment, etc., which does not address the concern regarding implementation in clinical practice. Thus, our ultimate goal remains, to establish this material in a clinical setting. We therefore sought to combine heat treatment and UV treatment to establish a nanostructured implant, while also investigating the initial responses of vascular endothelial cells, bone marrow cells and bacteria in the oral cavity immediately post implantation.

Surface properties of dental implants are a critical factor for achieving clinical success [33,34,35,36]. The topographical properties of nanostructures on titanium surface play a vital role in modulating cell response at the implant–tissue interface, which can significantly affect tissue integration into the implant [37,38,39]. To obtain solid initial fixation of the implant, it is necessary to accelerate the process of wound healing, promote cell adhesion and hard tissue differentiation, and inhibit bacterial growth. This study investigated the biocompatibility and antibacterial properties of heat-treated and UV-irradiated UV TNSs using RBM cells and Human Umbilical Vein Endothelial Cells (HUVECs).

In the present study, we investigated the properties of a TNS, formed by dark alkaline treatment heated at 600 °C followed by UV irradiation on angiogenesis, BM cell adhesion, induction of hard tissue differentiation, and inhibition of bacterial growth, to evaluate its suitability as a dental implant material.

## 2. Results

### 2.1. Surface Characterization

Scanning electron microscopy (SEM) analysis revealed a nanoporous network structure on the surface of the TNS disks (Figure 1a) similar to that described in a previous report [32]. TNS disks, treated by heating at 600 °C and then UV radiated (TNS-heat-UV), maintained this structure. The surface morphology and arithmetic mean roughness (Ra) of the TNS samples were examined by scanning probe microscopy (SPM), which showed the porous network structures (Figure 1b). Ra was not significantly altered by heat treatment and UV irradiation (TNS; 13.482 nm, TNS-heat; 14.281 nm, TNS-heat-UV; 13.285 nm).

Analysis of the TNS surface by X-ray photoelectron spectroscopy (XPS) revealed that the Ti2p and O1s peaks shifted in the heat-treated TNS (TNS-heat) and TNS-heat-UV relative to untreated TNS (Figure 1c). The C1s peak was reduced by UV and heat treatment of the TNS surface, indicating that short exposure to high-intensity UV radiation can decrease the surface carbon content of TNS. Figure 1d shows the thin-film X-ray powder diffractometry (TF-XRD) patterns of the NaOH-treated Ti surfaces exposed to heat and UV radiation; the main diffraction peaks were assigned according to a previous study [32]. The gel layer began to precipitate crystalline sodium titanate and rutile at approximately 600 °C. There were no differences in the TF-XRD peak distribution observed between TNS-heat and TNS-heat-UV. An analysis of water droplet contact-angles on the surface of Ti disks showed that alkali treatment improved the wettability of pure Ti metal, whereas heat treatment and UV irradiation resulted in superhydrophilicity (Figure 1e).

### 2.2. Protein Adsorption

The amount of bovine serum albumin (BSA) adsorbed on sample surfaces after 1, 3, 6, or 24 h of incubation was examined. TNS-heat-UV showed the highest amount of protein adsorption, which increased in a time-dependent manner (Figure 2).

### 2.3. In Vitro Tests Using RBM Cells

RBM cell adhesion on the TNS disks was assessed after 1, 3, 6, or 24 h of culture (Figure 3). Adherence was found to be highest in the TNS-heat-UV group at each time point. Phalloidin staining confirmed the adherence of cells and revealed the extension of cell processes in all groups, although the longest processes were observed in the TNS-heat-UV group. Next, we evaluated the osteogenic differentiation potential of RBM cells grown on the disks. After 7 and 14 days, alkaline phosphatase (ALP) activity was highest in the TNS-heat-UV group; this was accompanied by increased expression of osteocalcin (OCN) and higher calcium deposition relative to the TNS-heat and TNS control groups after 21 and 28 days. Additionally, osteogenesis-related genes, including ALP, runt-related transcription factor (Runx) 2, bone morphogenetic protein (BMP)-2, and osteopontin (OPN) were upregulated in a time-dependent manner in RBM cells grown on the different surfaces for 3, 7, 14, and 21 days, as determined by quantitative real-time (qRT-)PCR, with the highest levels observed in the TNS-heat-UV group (Figure 4).

### 2.4. In Vitro Tests Using Human Umbilical Vein Endothelial Cells (HUVECs)

Next, we examined the adhesion of HUVECs on the Ti disks after 0.5, 1, and 3 h of culturing and found that the TNS-heat-UV group showed the highest rate of adherence at each time point (Figure 5). Moreover, the mRNA levels of intercellular adhesion molecule (ICAM)-1 and Von Willebrand factor were higher in the TNS-heat-UV and TNS-heat groups compared to the control groups after 2 and 5 days of culture, respectively (*p* < 0.05 for both).

### 2.5. In Vitro Antibacterial Tests

The antibacterial potency of the TNS disks was evaluated using *Staphylococcus aureus*. The antibacterial rates were 0% for TNS-heat and 60% for TNS-heat-UV after 1 h, and 22% and 96%, respectively, after 6 h (Figure 6). These results indicate that TNS-heat-UV effectively killed the attached bacteria. Moreover, biofilm formation was observed on TNS and TNS-heat surfaces but not on TNS-heat-UV after 18 and 24 h (Figure 7).

We evaluated bacterial cell viability by live/dead staining and found that cells grown on TNS and TNS-heat rapidly proliferated within 6 h and showed 100% viability. In contrast, proliferation was markedly inhibited after 6 h in cells grown on TNS-heat-UV (Figure 8). To determine the cause of the decrease in cell viability, cells were stained with 2′,7′-dichlorodihydrofluorescein diacetate (DCFH-DA) to detect intracellular reactive oxygen species (ROS), which are generated by cells under redox stress and cause cell death. After 6 h of culture, cells grown on TNS and TNS-heat continued to proliferate until they covered the surface; DCFH-DA staining was negative. However, cells grown on TNS-heat-UV were positive for DCFH-DA after 1 and 6 h. These results demonstrate that TNS-heat-UV inhibits bacterial attachment, proliferation, and biofilm formation (Figure 8).

### 2.6. In Vivo Tests Using SD Rats

The reconstructed three-dimensional micro-computed tomography (CT) images of rat femur transverse slices, including implants, are shown in Figure 9. Both the test and control surface promoted new bone formation around the implants. More trabecular microarchitecture was observed near the TNS-heat-UV group than that of the TNS-modified titanium surface. Quantitative evaluation of the trabecular bone within the region of interest (ROI) is shown in Figure 10. Further, bone volume (BV)/tissue volume (TV), Tb.N, and Tb.Th were significantly higher in the test implant group compared to the control group (*p* < 0.05). Conversely, Tb.Sp exhibited a significantly lower value in the TNS-heat-UV group compared with the TNS group.

## 3. Discussion

Titanium modified by alkali treatment bonds well to bone, making it a useful material for clinical applications [24,25,26]. The TNS structure was generated by treatment with 10 M NaOH at 30 °C, and the SEM analysis showed that high heat treatment and UV irradiation did not damage the nanostructure. We previously demonstrated that this structure can regulate the osteogenic differentiation of RBM cells and enhance mineralization [24,25,26]. In the SPM analysis, the Ra of the titanium surface ranged from 13 to 14 nm and the nanostructure was similar to the hierarchical structure reported by other investigators. An Ra value between 13 and 16 nm was previously shown to be optimal for RBM cell culture [25,26].

We have reported that alkali-modified titanium surfaces promote new bone formation in vivo [30]. However, when using a larger animal, the torque applied at implant placement can destroy the nanostructure. Crystallization of the nanomaterial surface is considered necessary to generate a more stable structure [40]. Further, following oral surgery, the implant wound remains in direct contact with saliva, increasing the risk of bacterial contamination until host immune cells are recruited. UV treatment of the material surface can effectively reduce infection risk at the time of implantation [41,42,43]. The XRD revealed that sodium hydrogen titanate was gradually converted to amorphous sodium titanate and crystalline sodium titanate following heat treatment. Heating TNS-modified titanium to 500 °C caused the dehydration of the surface sodium titanate hydrogel layer; at 600 °C, this was transformed into crystallized amorphous sodium titanate [40]. The results of our study show that a temperature of 600 °C is optimal for inducing the formation of a crystalline nanostructure on the titanium surface. Photocatalytic activity strongly depends on surface redox potential; anatase with a larger band gap has a higher surface redox potential than rutile, and may therefore be a more effective photocatalyst. Other photo-inducible properties of TiO_2_ include high wettability—i.e., superhydrophilicity. Although this is distinct from photocatalytic bactericidal effects, both can occur simultaneously on the same TiO_2_ surface [44]. This improvement in wettability underlies the increased protein adsorption as well as angiogenic and osteogenic differentiation capacities of TNS-heat-UV compared to the other tested materials.

The early response of HUVECs is important for stable wound healing after implant surgery. Our results showed that HUVEC adhesion and growth were enhanced, relative to the control group, by heat treatment and UV irradiation of TNS, demonstrating that these processes can promote wound healing. Heat and UV radiation can function as growth factors for endothelial cells and play an essential role in the control of inflammation and revascularization. On the other hand, contact-angle, Ra, and surface energy influence the cell–substrate interaction [45]. Surface modification can also influence the activation of ALP [46], a marker of osteogenic differentiation. This study showed that rat bone marrow cells cultured on titanium surface in the TNS-heat-UV group show high ability to induce hard tissue differentiation. It is thought that chemical bonding through an apatite layer—such as between heat-treated Ti metal and bone following alkali treatment—influences osseointegration; apatite formation on an implant material surface is a precondition for this process [40]. In our study, the alkali- and heat-treated anatase of titanium formed apatite, which promoted RBM cell adhesion and osteogenic differentiation, and may effectively promote bone integration into implants.

In this study, TNS-heat-UV showed antibacterial activity, which is essential for the long-term success of dental implants. This effect could be related to the crystallinity of TiO_2_ induced by heat treatment. TiO_2_ is naturally polymorphic. By increasing the firing temperature, the crystal structure of TiO_2_ changes from amorphous to anatase and rutile. In addition to the crystal structure, the primary factor governing the photocatalytic antibacterial activity of TiO_2_ is surface area. When TiO_2_ is irradiated with UV light, the absorbed energy excites electrons in the valence shell, leaving holes (h^+^) that may be eliminated by recombination with additional electrons. Uncrystallized (amorphous) TiO_2_ does not exhibit photocatalytic antibacterial activity owing to a large number of lattice defects that trap e^−^ and h^+^ and serve as recombination centers. A higher degree of crystallinity reduces the number of lattice defects and increases the diffusion distance of e^−^ and H^+^, which is favorable for photocatalytic reactions [47,48,49,50]. Thus, the increased crystallization of TNS after heat treatment enhances the photocatalytic antibacterial activity. In addition, the conduction band gaps of anatase and rutile are approximately 3.2 and 3.0 eV, respectively; the higher value for anatase increases its oxidative degradation activity—and consequently, photocatalytic antibacterial activity—compared to rutile, even if the latter has higher crystallinity.

After UV irradiation, e^−^ that reach the TiO_2_ surface reduce oxygen from the atmosphere to form a superoxide anion (O_2_^−^), which has reducing power. Further reduction produces hydroperoxyl radical (HO_2_^•^), hydrogen peroxide (H_2_O_2_), and water. Additionally, H^+^ reaching the TiO_2_ surface react with hydroxide ions (OH^−^) in water or air to generate hydroxyl radicals (^•^OH), which have strong oxidizing power. O_2_^−^ and ^•^OH produced by UV irradiation of TiO_2_ can damage the cell wall and plasma membrane of bacteria, resulting in the leakage of intracellular substances and cell death [51].

Our in vitro results indicated that the TNS-heat-UV and TNS groups were appropriate for use in in vivo testing. The rat femur model in the present study was employed to evaluate the bone tissue response to implants via inclusion of trabecular bone alone, which is relevant in clinical settings. Healing, at this time point, is considered to be in the final stage in the rat models, whereas the degree of bone-implant contact is affected by the early bone response to the implant surface. In the present study, the bone-implant contact pattern for the TNS-heat-UV was enhanced, compared with that of the TNS-modified implants alone. Hence, Emdogain^®^-provided amelogenin may induce survival and differentiative signals in RBM cells, an effect which corresponds with its ascribed role in promoting early stages of in vivo bone formation, as reported by Kawana et al. [45]. This data supports our present study, and confirms that an amelogenin-coated TNS-modified titanium surface promotes bone differentiation around titanium implants not only in vitro but also in vivo. Further studies are necessary to investigate the mechanism of induction, utilized by the amelogenin-coated TNS-modified titanium surface.

## 4. Materials and Methods

### 4.1. Sample Preparation

Ti samples (JIS Grade 2, 15 mm in diameter) were punched from Ti sheets with a thickness of 1 mm (Daido Steel, Osaka, Japan). The disks were immersed in 10 M aqueous NaOH and maintained at 30 °C for 24 h. After alkali treatment, the samples were placed in an electrical furnace and heated to 600 °C at a rate of 5 °C/min under air atmosphere to induce crystallization. Crystalized TNS-modified disks were treated with ultraviolet light (wavelength = 254 nm, intensity = 100 mW/cm^2^) for 15 min using a UV irradiation instrument (HL-2000 HybriLinker; Funakoshi, Tokyo, Japan). Irradiated disks were divided into the following three groups: TNS-heat-UV (heat-treated and UV-irradiated Ti with nanonetwork structure), TNS-heat (heat-treated Ti with nanonetwork structure), and TNS (untreated Ti with nanonetwork structure).

### 4.2. Surface Characterization

The surface topography of TNS samples was qualitatively evaluated by SEM (S-4800 Shimadzu, Kyoto, Japan) and SPM (SPM-9600; Shimadzu). The composition of the coating was analyzed by XPS using a Kratos Analytical Axis Ultra DLD electron spectrometer (Kratos Instruments, Manchester, UK) with a monochromatic Al Kα X-ray source. Each sample was etched with Ar ions for 2 min (evaporation rate of 5 nm/min) to remove surface contaminants. The surface phase properties were investigated by TF-XRD (RINT-2500; Rigaku, Tokyo, Japan). Spectra were recorded in the range of 2θ = 20°–50°, operating at 50 kV and 300 mA, using a Cu-Kα radiation source, scanning speed of 2°/min, and incident angle of 1°. Contact-angle measurements were performed using a video contact-angle measurement system (VSA 2500 XE; AST Products, Tokyo, Japan) at room temperature. Ultrapure water was used for measurements.

### 4.3. Protein Adsorption Assay

BSA fraction V (Pierce Biotechnology, Rockford, IL, USA) was used as a model protein. Protein solution (300 µL, 1 mg/mL protein in saline) was pipetted onto each sample and after incubation for 1, 3, 6, or 24 h at 37 °C, non-adherent proteins were removed and mixed with bicinchoninic acid (Pierce Biotechnology) at 37 °C for 1 h. The removed and total amounts of inoculated BSA were quantified using a SpectraMax M5 microplate reader (Molecular Devices, Sunnyvale, CA, USA) at a wavelength of 562 nm. The rate of protein adsorption was calculated as the percentage of BSA adsorbed on samples relative to the total amount in solution.

### 4.4. Culture of RBM Cells

RBM cells were isolated from the femurs of 7-week-old Sprague-Dawley rats. The rats were euthanized using 4% isoflurane and the bones were aseptically excised from the hind limbs. The proximal end of the femur and the distal end of the tibia were clipped and a 21-gauge needle (Terumo, Tokyo, Japan) was inserted into the hole in the knee joint of each bone. The marrow was then flushed from the shaft with culture medium. The resultant marrow pellet was dispersed by trituration, and cell suspensions from all bones were combined in a centrifuge tube. RBM cells were cultured in 75 cm^2^ Falcon flasks (BD Biosciences, Franklin Lakes, NJ, USA) in culture medium. When they reached confluence, the cells were removed by trypsinization, washed twice with phosphate-buffered saline (PBS) (Dulbecco’s Modified Formula; ICN Biomedicals, Leicester, UK), resuspended in culture medium, and seeded at a density of 4 × 10^4^/cm^2^ in a 24-well Falcon tissue culture plate (BD Biosciences) containing titanium disks from all three groups. The cells were cultured at 37 °C in a humidified atmosphere of 5% CO_2_/95% air. This study was performed according to the Guidelines for Animal Experimentation of Osaka Dental University (approval no. 18-04008, 23 May, 2018).

### 4.5. In Vitro Tests Using RBM Cells

#### 4.5.1. Cell Viability

Cell proliferation was assessed with the CellTiter-Blue cell viability assay kit (Promega, Madison, WI, USA) according to the manufacturer’s protocol. Briefly, RBM cells were seeded on the TNS samples at a density of 4 × 10^4^/cm^2^ and allowed to adhere for 1, 3, 6, or 24 h; non-adherent cells were removed by rinsing with PBS. CellTiter-Blue reagent (50 µL) and PBS (250 µL) were added to each well. After incubation at 37 °C for 1 h, 100 µL of the solution was transferred to a new 96-well tissue culture plate, and the optical density at 560 and 590 nm was measured; the difference between the two values was defined as the proliferation value.

#### 4.5.2. Cell Morphology

To evaluate cell morphology, RBM cells were seeded on samples at a density of 4 × 10^4^/cm^2^. After 24 h, the adherent cells were washed with PBS, fixed with 4% paraformaldehyde solution for 20 min at room temperature, and permeabilized with 0.2% Triton X-100 for 30 min at room temperature. They were then incubated with Blocking One reagent (Nacalai Tesque, Kyoto, Japan) for 30 min at room temperature and stained with Alexa Fluor 488-phallodin (Invitrogen/Life Technologies, Carlsbad, CA, USA) and 4′,6-diamidino-2-phenylindole (DAPI) at 37 °C in the dark for 1 h. F-actin and cell nuclei were visualized by confocal laser scanning microscopy (LSM700; Carl Zeiss, Oberkochen, Germany).

#### 4.5.3. ALP Activity

To evaluate osteogenic differentiation based on ALP activity, RBM cells seeded on samples at a density of 4 × 10^4^/cm^2^ for 24 h were washed with PBS and lysed with 200 µL of 0.2% Triton X-100 (Sigma-Aldrich, St. Louis, MO, USA). After 7 or 14 days, the lysate was transferred to a microcentrifuge tube containing a 5 mm steel ball and agitated (Mixer Mill Type MM 301; Retsch GmbH, Haan, Germany) at 29 Hz for 20 s to homogenize the sample. ALP activity was measured with a luminometric enzyme-linked immunosorbent assay (ELISA) kit (Sigma-Aldrich) according to the manufacturer’s protocol. The reaction was terminated by adding 3 N NaOH at a final concentration of 0.5 N NaOH and measuring *p*-nitrophenol production using a microplate reader. DNA content was measured with the PicoGreen dsDNA assay kit (Invitrogen/Life Technologies) according to the manufacturer’s protocol, and the amount of ALP was normalized to the amount of DNA in the cell lysate.

#### 4.5.4. OCN Secretion

Osteogenic differentiation was evaluated by quantifying the amount of OCN secreted into the culture supernatant after 21 or 28 days by sandwich enzyme immunoassay using a commercial kit (Rat Osteocalcin ELISA Kit DS; DS Pharma Biomedical, Osaka, Japan) according to the manufacturer’s instructions.

#### 4.5.5. Mineralization Assay

Osteogenic differentiation was evaluated by detecting mineralization using a calcium E-test kit (Wako Pure Chemical Industries, Osaka, Japan). After 21 or 28 days of culture, 1 mL of calcium E-test reagent and 2 mL of buffer were added to 50 µL of collected medium. The absorbance of the reaction products was measured at 610 nm on a microplate reader. The concentration of calcium ions was calculated from the absorbance value relative to a standard curve.

#### 4.5.6. qRT-PCR

The expression of genes associated with osteogenic differentiation was evaluated by qRT-PCR. Total RNA was extracted from cells cultured for 3, 7, 14, and 21 days, and 1 µg was used to synthesize cDNA with the high-capacity cDNA archive kit (Applied Biosystems, Foster City, CA, USA). ALP, Runx2, BMP-2, and OPN mRNA levels were evaluated by qRT-PCR on a StepOne Plus Real-Time RT-PCR system (Applied Biosystems). A 10 µL volume of Taqman Fast Universal PCR Master Mix, 1 µL of the 20× Taqman gene expression assay primer probe set, 2 µL of sample cDNA, and 7 µL of diethylpyrocarbonate-treated water (Nippongene, Toyama, Japan) were added to each well of a MicroAmp Fast Optical 96-well microplate (0.1-mL well volume; Applied Biosystems). The amplification reaction consisted of 40 cycles of 95 °C for 1 s and 60 °C for 20 s. Target gene expression levels were calculated relative to the negative control group with the 2^−ΔΔ*C*t^ method.

### 4.6. Culture of HUVECs

HUVECs were purchased from CellWorks (Buckingham, UK) and cultured in endothelial cell growth medium (HuMedia-MvG; Kurabou, Osaka, Japan) supplemented with 5% fetal bovine serum, 10 ng/mL recombinant human epithelial growth factor, 1 μg/mL hydrocortisone hemisuccinate, 50 μg/mL gentamicin, 5 ng/mL amphotericin B, 5 ng/mL recombinant basic human fibroblastic growth factor, 10 μg/mL heparin, and 39.3 μg/mL dibutyryl-cyclic AMP (Kurabou) in a 75 cm^2^ culture flask coated with type I collagen (Asahi Technoglass, Tokyo, Japan). Cells were washed three times with washing buffer, mounted with CD31 reagent, and photographed under a microscope equipped with a digital camera to characterize cell morphology.

### 4.7. In Vitro Tests Using HUVECs

#### 4.7.1. Cell Viability

HUVECs were seeded on TNS samples at a density of 4 × 10^4^/cm^2^ for 3 h to allow cell attachment. Cell adhesion was measured after 0.5, 1, and 3 h with the CellTiter-Blue cell viability assay kit (Promega) according to the manufacturer’s protocol.

#### 4.7.2. Cell Morphology

HUVECs were seeded on TNS samples at a density of 4 × 10^4^/cm^2^ for 3 h to allow cell attachment. They were then incubated with Blocking One reagent for 30 min at room temperature and stained with Alexa Fluor 488-phallodin and DAPI at 37 °C in the dark for 1 h. F-actin and cell nuclei were visualized by confocal laser scanning microscopy.

#### 4.7.3. qRT-PCR

The expression of genes associated with angiogenesis including ICAM-1 (after 3 days of culture) and von Willebrand factor and thrombomodulin (after 7 days of culture) was evaluated by qRT-PCR according to the protocol described in Section 4.5.6.

### 4.8. Bacterial Cultivation

*S. aureus* (ATCC 12600; American Type Culture Collection, Manassas, VA, USA) was streaked on trypticase soy agar and incubated at 37 °C for 24 h under aerobic conditions. A single colony was selected and cultivated overnight in 5 mL of trypticase soy broth. This seed culture was diluted in fresh trypticase soy broth, and 20 μL of the bacterial suspension (10^6^ CFU) were seeded on disks in a standard 12-well polystyrene culture plate and incubated aerobically at 37 °C for 24 h.

### 4.9. Antibacterial Activity Assay

After 1 and 6 h of incubation, bacteria that had adhered to the disks were collected in 2 mL of fresh broth by rapid vortexing for 1 min at maximum power; the cell suspension was serially diluted and spread on trypticase soy agar plates and cultured for 24 h. The bacterial colonies were counted and growth inhibition was calculated as % = (A − B)/A × 100%, where A is the average number of bacteria on TNS and B is the average number of bacteria on TNS-heat and TNS-heat-UV (all in CFU).

### 4.10. Evaluation of Bacterial Cell Viability by Live/Dead Staining

The bacterial suspension was removed after 1 and 6 h of incubation, and the disks were transferred to a new 12-well polystyrene culture plate, rinsed with sterile PBS to dislodge and remove unattached bacteria, and stained for 15 min using the Live/Dead BacLight bacterial viability kit (Thermo Fisher Scientific, Waltham, MA, USA; cat. no. L7012) according to the manufacturer’s instructions. Excess dye was removed and the disks were immediately photographed on a cover glass with a confocal laser scanning microscope.

### 4.11. Biofilm Formation Assay

After 18 and 24 h of incubation, the disks were gently rinsed with PBS and incubated for 20 min at room temperature with 2 mL of 0.05% *w*/*v* crystal violet dye. After rinsing with PBS to remove residual dye, the disks were transferred to a 24-well plate and destained for 20 min at room temperature by rotary shaking in 1 mL of 95% ethanol. A volume of 100 µL of ethanol was added to each well and the absorbance at 595 nm was measured on a microplate reader.

### 4.12. Measurement of ROS Level

Intracellular ROS levels were investigated by fluorescence imaging. After incubation for 1 and 6 h, the disks were washed with PBS, and 300 μL of DCFH-DA (Sigma-Aldrich) was added to the disk surface, followed by incubation at 37 °C for 30 min. Excess dye was removed by washing with PBS and the samples were examined with a confocal laser scanning microscope.

### 4.13. Statistical Analysis

Experiments were performed in triplicate. Data are presented as the mean ± standard deviation. Differences between groups were evaluated by one-way analysis of variance followed by the Tukey test, and *p* < 0.05 was considered statistically significant.

### 4.14. In Vivo Test Using SD Rats

#### 4.14.1. Animal Model and Surgical Procedures

A total of 20 male SD rats (Shimizu Laboratory Supplies Co., Kyoto, Japan; age 8 weeks, weighing 160 ± 15 g) were used in this study. The experimental animals were randomly divided into two groups (TNS-heat-UV and TNS) based on the in vitro test results, with 10 rats in each group. The experimental method was based on previous studies our research group undertook. Animals were administered inhalation anesthesia, followed by intraperitoneal injection of anesthetics (1.5 mL/kg). The fur was shaved from the right hind limb and the skin was disinfected with iodine followed by 75% ethanol to remove the iodine. A 1 cm long longitudinal skin incision was made along the medial side of the knee joint and the subcutaneous fascia was incised. The patella and extensor mechanism were then dislocated to expose the distal aspect of the femur. A pilot hole was drilled through the intercondylar notch using a 1 mm round dental burr under profuse sterile saline irrigation The hole was enlarged to 1.2 mm with an endodontic file. The implants, sterilized by ethylene oxide gas, were randomly inserted into the 20 prepared channels and the medullary cavities of the right femurs. After surgery, the knee joint was restored and the surgical site was closed in layers. The animals received intramuscular injections of gentamicin (1 mg/kg) and buprenorphine (0.05 mg/kg) for 3 days to prevent postsurgical infection and relieve pain. All rats were allowed free movement without any restriction. 

#### 4.14.2. Micro-CT Analysis

Immediately after dissection, the right femurs including the implants were placed in cool saline solution and scanned with an SMX-130CT micro-computed tomography (micro-CT) scanner (Shimadzu) operated at 70 kV and 118 mA; the isotropic voxel size was 10 μm in all spatial directions. After the tomographic acquisitions, the implant and surrounding tissues were reconstructed and analyzed using morphometric software (TRI/3D-BON; Ratoc System Engineering, Tokyo, Japan). The ROI was defined as the 500 μm wide area of bone around the implants from 2 mm below the highest point of the growth plate to the distal 100 slices [32]. The bone volume fraction (BV/TV), mean trabecular number (Tb.N), mean trabecular thickness (Tb.Th), and mean trabecular separation (Tb.Sp) were calculated within the ROI.

## 5. Conclusions

The results of this study demonstrate that the combination of heating and UV irradiation increased the biocompatibility and antibacterial properties of a Ti-based material. The treatment enhanced the angiogenic and osteogenic induction capacities of TNS, as well as the adherence of HUVECs and RBM cells, while reducing the viability of the oral pathogen *S. aureus* on the material surface by stimulating ROS production. Thus, heat-treated, UV-irradiated TNS is a promising material for the fabrication of dental, and possibly other types of implants.

## Figures and Tables

**Figure 1 ijms-20-05991-f001:**
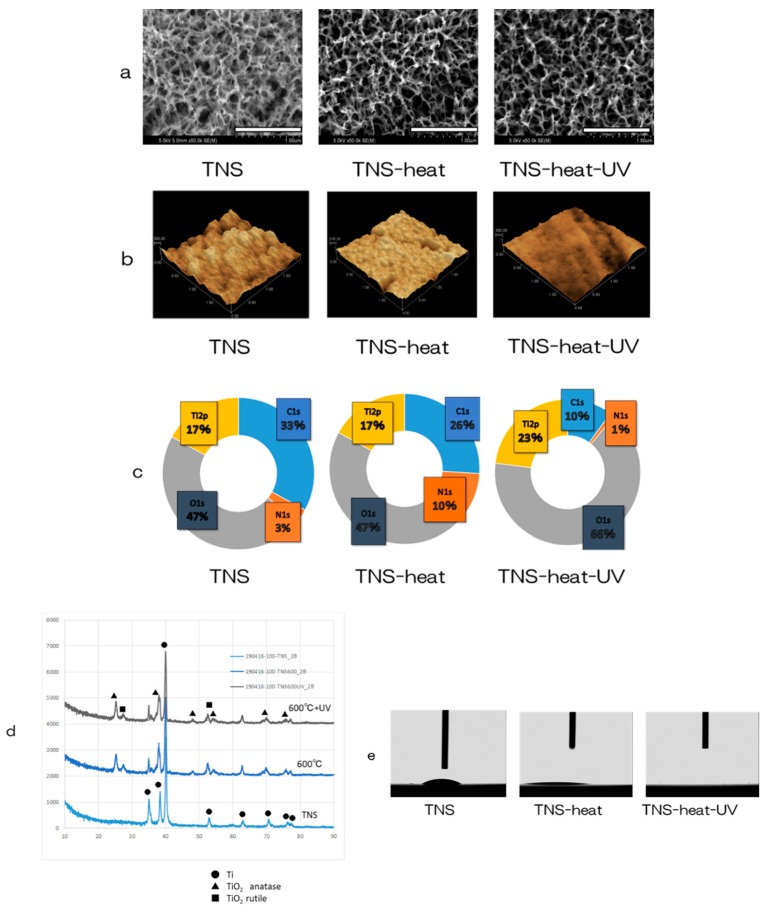
Surface analysis; (**a**) Scanning electron microscopy (SEM), (**b**) scanning probe microscopy (SPM), (**c**) X-ray photoelectron spectroscopy (XPS), (**d**) thin-film X-ray powder diffractometry (TF-XRD), (**e**) contact-angle.

**Figure 2 ijms-20-05991-f002:**
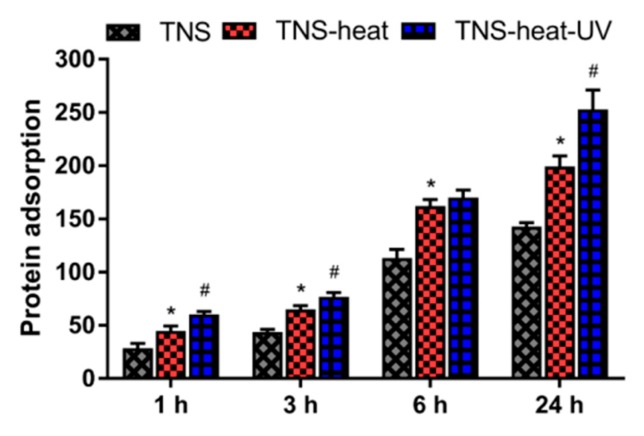
The amount of bovine serum albumin (BSA) adsorbed on sample surfaces after 1, 3, 6, or 24 h of incubation was examined. TNS (Ti-nanosheet)-heat-UV showed the highest amount of protein adsorption, which increased in a time-dependent manner (* *p* < 0.05 vs. TNS, # *p* < 0.05 vs. TNS-heat).

**Figure 3 ijms-20-05991-f003:**
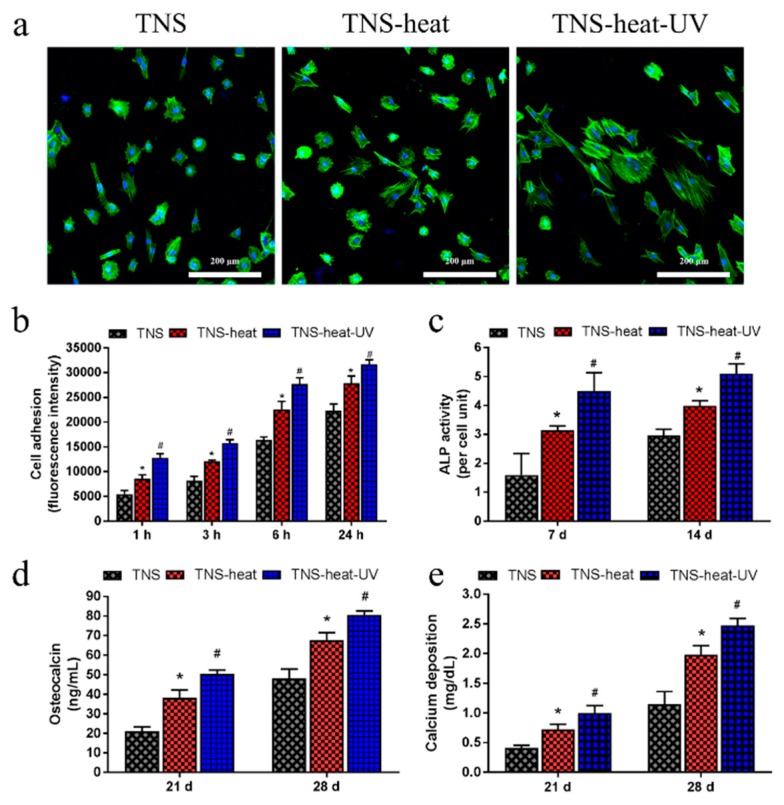
(**a**) Phalloidin staining confirmed the adherence of cells and revealed the extension of cell processes in all groups, although the longest processes were observed in the TNS-heat-UV group; (**b**) Rat bone marrow (RBM) cell adhesion on the TNS disks was assessed after 1, 3, 6, or 24 h of culture. Adherence was highest in the TNS-heat-UV group at each time point; (**c**) After 7 and 14 days, alkaline phosphatase (ALP) activity was highest in the TNS-heat-UV group. (**d**,**e**) Further, an increase in the expression of osteocalcin (OCN), together with higher calcium deposition was observed in the TNS-heat-UV group relative to the TNS-heat and TNS control groups after 21 and 28 days (* *p* < 0.05 vs. TNS, # *p* < 0.05 vs. TNS-heat).

**Figure 4 ijms-20-05991-f004:**
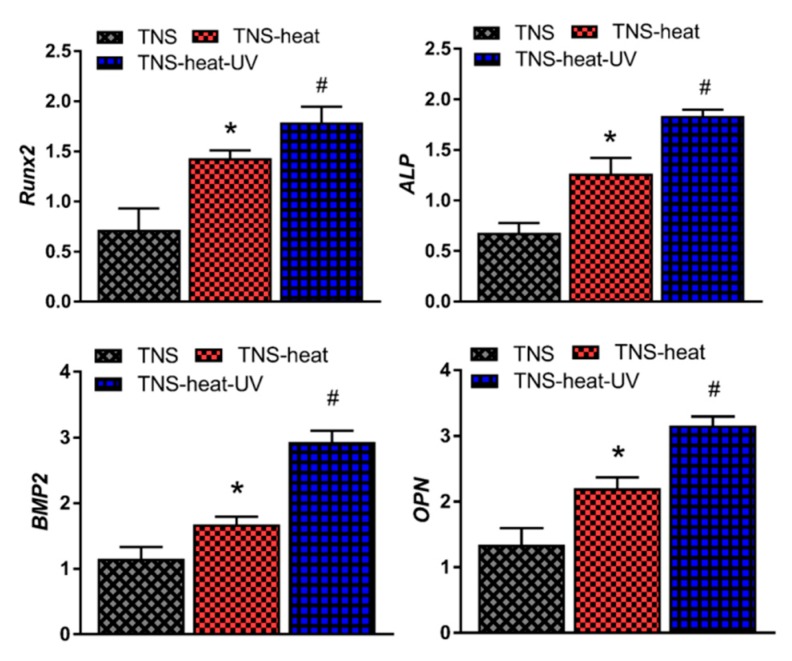
Osteogenesis-related genes including ALP, runt-related transcription factor (Runx2), bone morphogenetic protein (BMP)-2, and osteopontin (OPN) were upregulated in a time-dependent manner in RBM cells grown on different surfaces for 3, 7, 14, and 21 days, as determined by quantitative real-time PCR (qRT- PCR), with the highest levels observed in the TNS-heat-UV group (* *p* < 0.05 vs. TNS, # *p* < 0.05 vs. TNS-heat).

**Figure 5 ijms-20-05991-f005:**
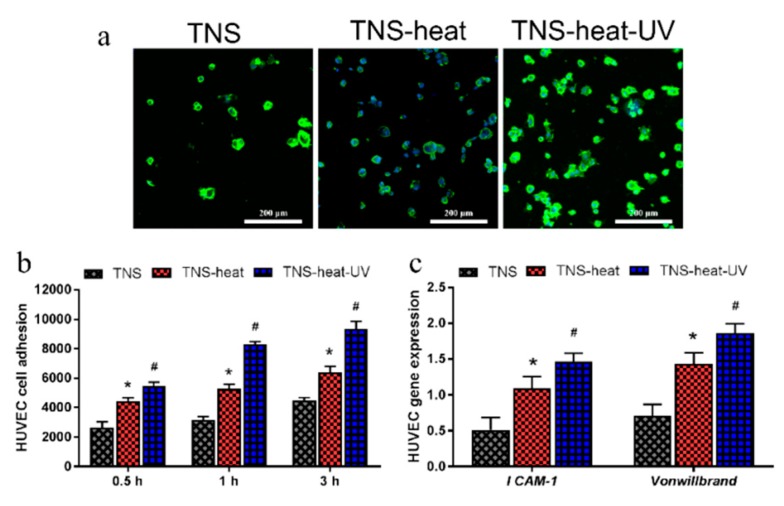
(**a**) TNS-heat-UV group showed the highest rate of adherence at each time point; (**b**) The adhesion of Human Umbilical Vein Endothelial Cells (HUVECs) on the Ti disks after 0.5, 1, and 3 h of culture; (**c**) The mRNA levels of intercellular adhesion molecule (ICAM)-1 and Von Willebrand factor were higher in the TNS-heat-UV and TNS-heat groups than in the control groups after 2 and 5 days of culture, respectively (* *p* < 0.05 vs. TNS, # *p* < 0.05 vs. TNS-heat).

**Figure 6 ijms-20-05991-f006:**
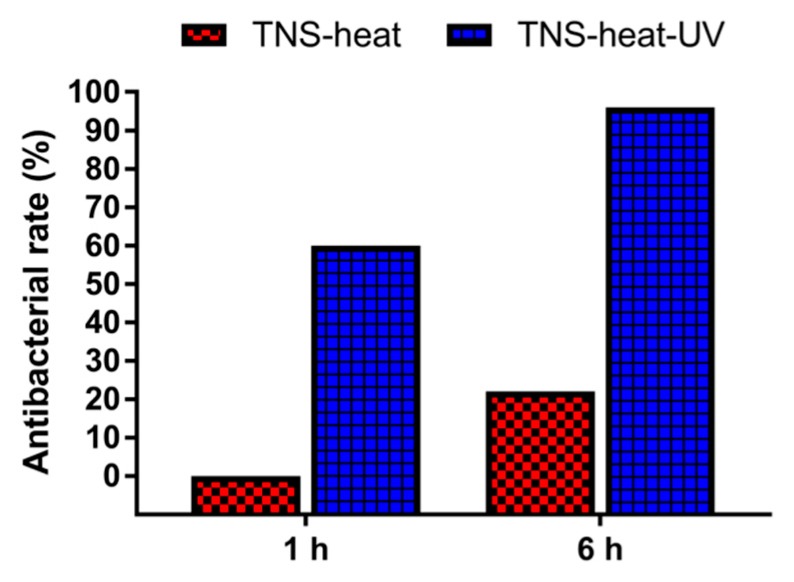
The antibacterial potency of the TNS disks was evaluated using *Staphylococcus aureus*. The antibacterial rates were 0% for TNS-heat and 60% for TNS-heat-UV after 1 h, and 22% and 96%, respectively, after 6 h.

**Figure 7 ijms-20-05991-f007:**
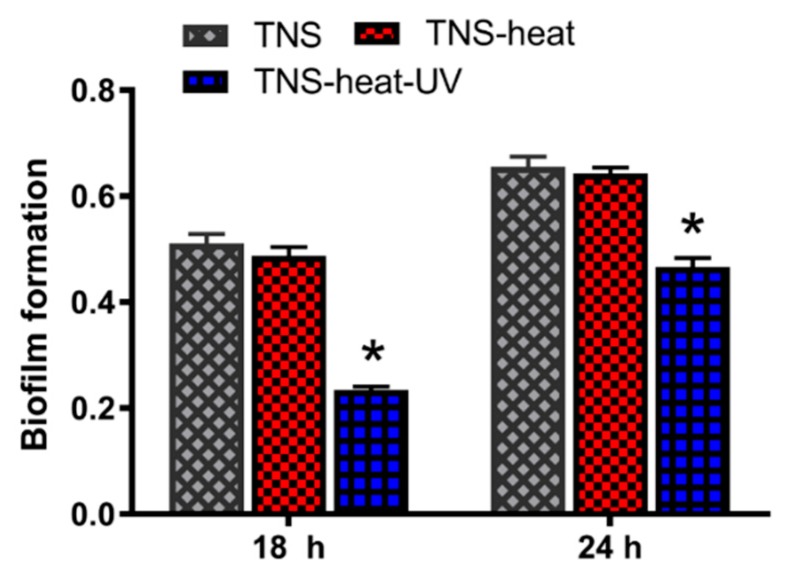
TNS, TNS-heat, and TNS-heat-UV disks were incubated with *Staphylococcus aureus* MG-1. Biofilm formation was evaluated by crystal violet staining after 18 and 24 h incubation. Biofilm formation was observed on TNS and TNS-heat but fewer on TNS-heat-UV after 18 and 24 h. (* *p* < 0.05 vs. TNS)

**Figure 8 ijms-20-05991-f008:**
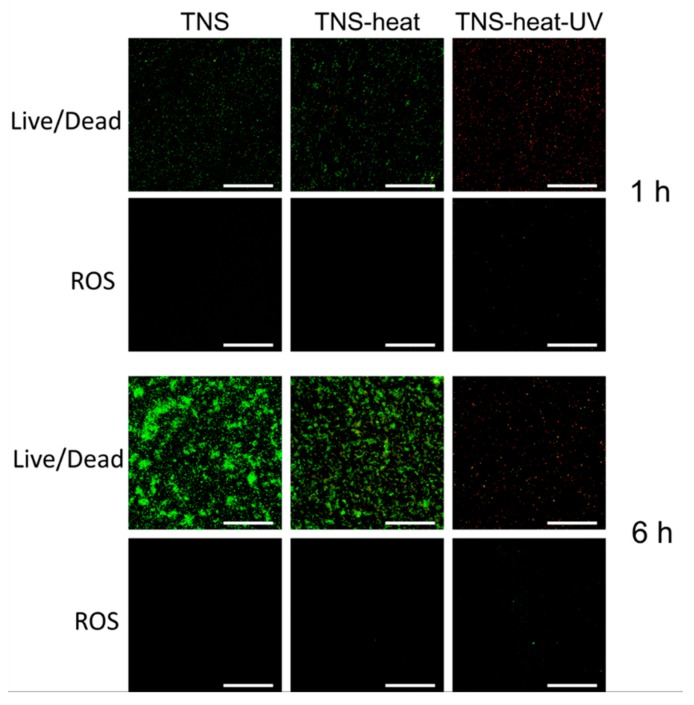
Live/dead and Reactive Oxygen Species (ROS) staining of the bacteria attached to sample disks. After 6 h of culture, bacteria grown on TNS and TNS-heat continued to proliferate until they covered the surface; 2′,7′-dichlorodihydrofluorescein diacetate (DCFH-DA) staining was negative (Figure 6). However, bacteria grown on TNS-heat-UV were positive for DCFH-DA after 1 and 6 h. Scale bar = 200 µm.

**Figure 9 ijms-20-05991-f009:**
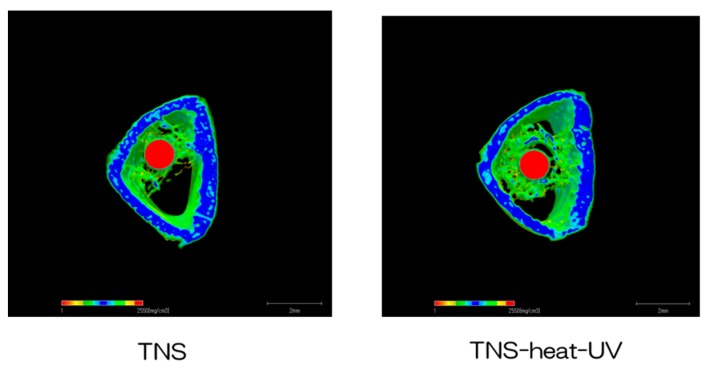
Micro-CT images (the implants were marked with red color, the cortical bone with blue color, and the cancellous bone with green color).

**Figure 10 ijms-20-05991-f010:**
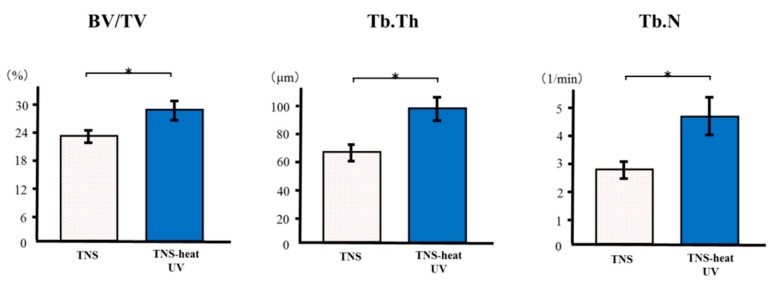
Quantitative evaluation of the trabecular bone within ROI (region of interest) (BV/TV, Tb.N, Tb.Th and Tb.Sp) * *p* > 0.05.

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
