# Peer review of "UV Treatment Improves the Biocompatibility and Antibacterial Properties of Crystallized Nanostructured Titanium Surface"

_ijms, 2019, doi:10.3390/ijms20235991_

Round 1
Reviewer 1 Report
The authors present a study investigating the impact of UV and heat treatment on anti-bacterial performance as evaluated using bone marrow and endothelial cell adherence. The manuscript was well presented with a logical approach to experimental design and discussion. The authors showed reduced bacterial adherence and improved performance of the UV titanium. They discussed the potential advantages for future dental implant design.
The paper had very few typographical errors and was written in a clear concise manner. The results were clearly presented and reflected a wide range of methods used during the study. The method section gave sufficient detail in order to understand the experiential design with appropriate statistical analysis.
The conclusions appear to be in keeping with the findings.
Author Response
Thank you very much for circulating our manuscript entitled “UV treatment improves the biocompatibility and antibacterial properties of crystallized nanostructured titanium surface” by Mai Hatoko et al. among the members of the editorial board of the International Journal of Molecular Science and forwarding two reviewers’ suggestions to me. I have enclosed our responses to the reviewers’ comments. Attached are ten files associated with our manuscript.
Reviewer #1:
Thank you very much for your comments. We have revised our manuscript in accordance with your suggestions.
Reviewer 2 Report
Manuscript ID: IJMS-638126
Title: UV treatment improves the biocompatibility and antibacterial properties of crystallized nanostructured titanium surface
In this work, the authors fabricated the Titanium Nanosheets (TNS) having a crystalline nanostructure on Pure Titanium metal and investigated and compared biocompatibility and antibacterial properties of heat treated and UV irradiated TNS against the untreated TNS. In vitro and in vivo experiments were performed to assess biocompatibility and effects on cell behaviour using human umbilical vein endothelial cells and rat bone marrow cells. Overall, the manuscript is well written and the work performed is novel. However, authors need to address following issues before the work can be accepted for the publication in IJMS.
C#1: Line 70: “Thus, our ultimate goal remains, to establish this material in a clinical setting,.” Remove the comma after “setting,”.
C#2: In literature review, for better understanding, authors need to discuss the reasons why they selected heating temperature of 600°C with citing the suitable reference. What is effect of temperature of structure formed?
C#3: Section 2.1: The provided SEM images of TNS disks are not readable. Authors should present the Figure 1 in appropriate way as the most part of figure 1 is not readable including the axes and units. It is claimed by authors that the TNS disk reveals nanoporous network structure on the surface and this is similar to that described in a previous report [33]. However, the reference [33] seems to be incorrect instead of [32].
C#4: Line 80: “Scanning electron microscopy (SEM) analysis revealed a nanoporous network structure on the…”. Authors should to disclose the scientific reasons of such nanostructure formation, briefly. Moreover, authors need to compare and discuss effect of treatment method on the nanostructure of TNS-Heat and TNS-heat-UV samples with that of untreated TNS.
C#5: Line 84: “Ra was not significantly altered by heat treatment and UV irradiation.(TNS; 13.482, TNS-heat; 14.281…”. Full stop before the bracket should be removed and placed after the bracket. The unit of Ra value should be included.
C#6: Authors need to provide and compare the contact angle values for samples and discuss the reasons behind trends. It is claimed that the heat treatment and UV irradiation does not changes the surface roughness significantly. Then, why does they affect the contact angle change?
C#7: 3. Discussion
Initial 2-3 paragraphs of discussion section looks like the literature review and authors can remove that from discuss section and put in section1. In discussion, they should focus to deliberate the scientific discussion behind the observed results.
C#8: Line 201: “In the SPM analysis, the Ra of the titanium surface ranged from 17 to 19 nm…”. However, the mentioned values of Ra in section 2.1 are ranging between 13 to 14 nm. Kindly check and correct accordingly.
C#9: Line 214 “The results of our study show that a temperature of 600°C is optimal for inducing the formation of a crystalline…”. Authors claim that in this work they found the temperature of 600°C to be optimum for inducing the formation of a crystalline structure. However, there is not such comparison of effect of heating temperature on the structure of the TNS.
C#10: The nomenclature of sample groups is not consistent throughout the paper, e.g. the TNS-heat-UV is mentioned as TNS-600-UV at some places and so on. Kindly make sure consistent nomenclature of samples throughout the manuscript.
C#11: Line 230: “Indeed, our results showed that RBM..”. Rewrite the sentence for proper understanding.
Author Response
Thank you very much for circulating our manuscript entitled “UV treatment improves the biocompatibility and antibacterial properties of crystallized nanostructured titanium surface” by Mai Hatoko et al. among the members of the editorial board of the International Journal of Molecular Science and forwarding two reviewers’ suggestions to me. I have enclosed our responses to the reviewers’ comments. Attached are ten files associated with our manuscript.
C#1: Line 70: “Thus, our ultimate goal remains, to establish this material in a clinical setting,.” Remove the comma after “setting,”.
We agree with your suggestion and have corrected our manuscript as follows:
Page 2, lines 67-70:
However, these previous studies have been shown to be specific to an individual variable such as heat treatment, UV treatment, etc., which does not address the concern regarding implementation in clinical practice. Thus, our ultimate goal remains, to establish this material in a clinical setting.
C#2: In literature review, for better understanding, authors need to discuss the reasons why they selected heating temperature of 600°C with citing the suitable reference. What is effect of temperature of structure formed?
Page 2 Line 61-65
In recent study, we examined how the surface of TNS modified Ti and Ti4Al6V alloy is affected by heat treatment at multiple temperatures. As a result, heat treatment at 600 ° C. formed a TNS crystal structure in which an amorphous layer of alkali titanate was firmly bonded to the substrate on the surface of Ti and Ti4Al6V alloy.
C#3: Section 2.1: The provided SEM images of TNS disks are not readable. Authors should present the Figure 1 in appropriate way as the most part of figure 1 is not readable including the axes and units. It is claimed by authors that the TNS disk reveals nanoporous network structure on the surface and this is similar to that described in a previous report [33]. However, the reference [33] seems to be incorrect instead of [32].
We agree with your suggestion and have corrected figure1.
C#4: Line 80: “Scanning electron microscopy (SEM) analysis revealed a nanoporous network structure on the…”. Authors should to disclose the scientific reasons of such nanostructure formation, briefly. Moreover, authors need to compare and discuss effect of treatment method on the nanostructure of TNS-Heat and TNS-heat-UV samples with that of untreated TNS.
Thank you for your advice. Our previous report revealed that nano-structures containing sodium titanate are formed on the material surface by treating titanium surface with alkali treatment. This study shows that there is no change in the mechanical structure of the material due to heat treatment or UV treatment, but a chemical change.
C#5: Line 84: “Ra was not significantly altered by heat treatment and UV irradiation.(TNS; 13.482, TNS-heat; 14.281…”. Full stop before the bracket should be removed and placed after the bracket. The unit of Ra value should be included.
We agree with your suggestion and have corrected our manuscript as follows:
Page 2, lines 85-86:
TNS; 13.482 nm, TNS-heat; 14.281 nm, TNS-heat-UV13.285 nm
C#6: Authors need to provide and compare the contact angle values for samples and discuss the reasons behind trends. It is claimed that the heat treatment and UV irradiation does not changes the surface roughness significantly. Then, why does they affect the contact angle change?
Thank you for your advice. The reason for the decrease in contact angle is not only due to surface roughness, but also due to changes in the surface potential of the material surface and chemical changes. In this study, it is considered that the alkali treatment induced a large decrease in the contact angle, and the material surface was crystallized and UV treatment was added to show super hydrophilicity.
C#7: 3. Discussion
Initial 2-3 paragraphs of discussion section looks like the literature review and authors can remove that from discuss section and put in section1. In discussion, they should focus to deliberate the scientific discussion behind the observed results.
We agree with your suggestion and have corrected our manuscript.
C#8: Line 201: “In the SPM analysis, the Ra of the titanium surface ranged from 17 to 19 nm…”. However, the mentioned values of Ra in section 2.1 are ranging between 13 to 14 nm. Kindly check and correct accordingly.
We agree with your suggestion and have corrected our manuscript as follows:
Page 8, lines 206-208:
We previously demonstrated that this structure can regulate the osteogenic differentiation of RBM cells and enhance mineralization [24–26]. In the SPM analysis, the Ra of the titanium surface ranged from 13 to 14 nm,
C#9: Line 214 “The results of our study show that a temperature of 600°C is optimal for inducing the formation of a crystalline…”. Authors claim that in this work they found the temperature of 600°C to be optimum for inducing the formation of a crystalline structure. However, there is not such comparison of effect of heating temperature on the structure of the TNS.
Thank you for your advice. We add this sentence with our manuscript
Page 2 Line 61-65
In recent study, we examined how the surface of TNS modified Ti and Ti4Al6V alloy is affected by heat treatment at multiple temperatures. As a result, heat treatment at 600 ° C. formed a TNS crystal structure in which an amorphous layer of alkali titanate was firmly bonded to the substrate on the surface of Ti and Ti4Al6V alloy.
C#10: The nomenclature of sample groups is not consistent throughout the paper, e.g. the TNS-heat-UV is mentioned as TNS-600-UV at some places and so on. Kindly make sure consistent nomenclature of samples throughout the manuscript.
We agree with your suggestion and have corrected our manuscript.
C#11: Line 230: “Indeed, our results showed that RBM..”. Rewrite the sentence for proper understanding.
Thank you for your advice. We add this sentence with our manuscript
Page 9 Line 236-237
This study showed that rat bone marrow cells cultured on titanium surface in the TNS-heat-UV group show high ability to induce hard tissue differentiation.
Round 2
Reviewer 2 Report
Manuscript ID: IJMS-638126
Title: UV treatment improves the biocompatibility and antibacterial properties of crystallized nanostructured titanium surface
C#1: Figures 3 and 4: units not clear on the y axis for example how is 'cell adhesion' (Fig.3a) measured, these units don't match the description in the methods. For the ALP data (Fig.3b) this looks like some kind of relative units please elaborate in the axis or legend (the methods states ALP/DNA but this should be expressed as per micrograms of DNA or per DNA/minute. In general these legends are written more in the style of the results section text whereas they could be less wordy but with more technical information.
C#2: Figures 7 and 8 legends are also too short and do not have enough technical details.
Author Response
C#1: Figures 3 and 4: units not clear on the y axis for example how is 'cell adhesion' (Fig.3a) measured, these units don't match the description in the methods. For the ALP data (Fig.3b) this looks like some kind of relative units please elaborate in the axis or legend (the methods states ALP/DNA but this should be expressed as per micrograms of DNA or per DNA/minute. In general these legends are written more in the style of the results section text whereas they could be less wordy but with more technical information.
We agree with your suggestion and have corrected Figure 3 and 4.
C#2: Figures 7 and 8 legends are also too short and do not have enough technical details.
We agree with your suggestion and have corrected our manuscript as follows:
Page 6, lines 170-172:
TNS, TNS-heat, and TNS-heat-UV disks were incubated with Staphylococcus aureus MG-1. Biofilm formation was evaluated by crystal violet staining after 18 and 24 h incubation.
Page 7, lines 183-186
Live/dead and ROS staining of the bacteria attached to sample disks. After 6 h of culture, cells grown on TNS and TNS-heat continued to proliferate until they covered the surface; DCFH-DA staining was negative (Fig. 6). However, cells grown on TNS-heat-UV were positive for DCFH-DA after 1 and 6 h. Scale bar=200 µm.
This manuscript is a resubmission of an earlier submission. The following is a list of the peer review reports and author responses from that submission.
Round 1
Reviewer 1 Report
Dear authors,
your study is well prepared. There is only one remark. When an antibacterial effect is determined it is better to use reduction in CFU in log. If possible please change from estimation in % to count in log. A measure can be called antimicrobial active if there is a reduction in CFU by at least 3 log counts. Please consider and discuss throughout your study.
Reviewer 2 Report
The manuscript is well written, but I hope in vivo data in this manuscript.
A lot of surface study concerning dental implant are published, but now in the mainstream, in vitro and in vivo data is the set for good manuscript.
The reader cannot believe the data only with in vitro data, now.